# Sex Differences in Parkinson’s Disease: From Bench to Bedside

**DOI:** 10.3390/brainsci12070917

**Published:** 2022-07-13

**Authors:** Maria Claudia Russillo, Valentina Andreozzi, Roberto Erro, Marina Picillo, Marianna Amboni, Sofia Cuoco, Paolo Barone, Maria Teresa Pellecchia

**Affiliations:** Center for Neurodegenerative Diseases (CEMAND), Department of Medicine, Surgery and Odontology “Scuola Medica Salernitana”, University of Salerno, 84084 Fisciano, Italy; mariaclaudiarussillo93@gmail.com (M.C.R.); valentinandreozzi@gmail.com (V.A.); rerro@unisa.it (R.E.); mpicillo@unisa.it (M.P.); mamboni@unisa.it (M.A.); scuoco@unisa.it (S.C.); pbarone@unisa.it (P.B.)

**Keywords:** Parkinson’s disease, gender, sex

## Abstract

Background: Parkinson’s disease (PD) is the second most common neurodegenerative disorder after Alzheimer’s disease and gender differences have been described on several aspects of PD. In the present commentary, we aimed to collect and discuss the currently available evidence on gender differences in PD regarding biomarkers, genetic factors, motor and non-motor symptoms, therapeutic management (including pharmacological and surgical treatment) as well as preclinical studies. Methods: A systematic literature review was performed by searching the Pubmed and Scopus databases with the search strings “biomarkers”, “deep brain stimulation”, “female”, “gender”, “genetic”, “levodopa”, “men”, “male”, “motor symptoms”, “non-motor symptoms”, “Parkinson disease”, “sex”, “surgery”, and “women”. Results: The present review confirms the existence of differences between men and women in Parkinson Disease, pointing out new information regarding evidence from animal models, genetic factors, biomarkers, clinical features and pharmacological and surgical treatment. Conclusions: The overall goal is to acquire new informations about sex and gender differences in Parkinson Disease, in order to develop tailored intervetions.

## 1. Introduction

It has been known for many years that sex affects morphology and function of the brain; consequently, specific sex-related factors contribute to determining phenotypic differences even in neurodegenerative diseases. Parkinson’s disease (PD) is the second most common neurodegenerative disorder after Alzheimer’s disease and gender differences have been described on several aspects of PD.

In the present review, we aim to present the currently available evidence on gender differences in PD regarding biomarkers, genetic factors, motor and non-motor symptoms, therapeutic management (including pharmacological and surgical treatment) as well as preclinical studies, starting from 2017’s **“The relevance of gender in Parkinson disease: a review”** by Picillo et al. and collecting all the new evidence emerged since then [1].

## 2. Methods

We searched PubMed for peer-reviewed articles published in English from 2017 to present; the most recent of the included publications dates back to 1 June 2022. The search terms “biomarkers”, “deep brain stimulation”, “female”, “gender”, “genetic”, “levodopa”, “men”, “male”, “motor symptoms”, “non-motor symptoms”, “Parkinson disease”, “sex”, “surgery”, and “women” were used. Additional articles were identified by searching the reference lists of identified reviews. Articles are summerized in Table 1.

## 3. Results

### 3.1. Evidence from Animal Models

Preclinical studies often did not consider sex as a variable, so we aim to review evidence of sex differences in PD animal models.

Gender differences in neurotoxicity have been reported, and several research in experimental animals indicates that estrogen protects dopaminergic neurons from various types of toxic injury.

Studies in animal models have shown a greater neurotoxic effect in male than female mice, with a higher depletion of striatal dopamine (DA) in male mice treated with MPTP (1-metyl-4-fenyl-1,2,3,6-tetrahydropyridin) and a major reduction of dopamine transported (DAT) specific binding in the substantia nigra in male mice treated with methamphetamine [63,64,65,66]. Interestingly, when animals were treated with estrogen prior to neurotoxin administration, higher striatal DA concentrations were observed [67,68], suggesting that estrogen may have a role in neuroprotection of DA neurons.

However, the role of estrogens is still unclear, and some data are non consistent with their neuroprotective effect. In a recent study on a rat model of progressive parkinsonism obtained with repeated administrations of low doses of reserpine, ovariectomized female rats presented a lower susceptibility to the effect of reserpine, when compared with male and female intact rats, a result of unclear interpretation. However, overall, female rats presented a lower susceptibility to the deleterious effects of reserpine on DA neurons [69,70].

The MitoPark (MP) mouse, a transgenic mitochondrial impairment model, recapitulates key features of human PD with loss of substantia nigra neurons, depletion of DA in the striatum, loss of voluntary movements, tremor, responsiveness to L-DOPA treatment, so this model spontaneously exhibits progressive motor deficits and neurodegeneration [71]. An advantage of this model is the elimination of the additional pharmacological effects of neurotoxins in DA neurons and non-DA cell types. Studies on gender differences using this model confirmed results obtained with other models, with male MP mice being more affected than female MP mice, and ovariectomized female MP mice showing a similar time course as that of male MP mice [72].

Prodopaminergic or antidopaminergic effects of androgens received less attention than estrogens, but an important question is to determine whether there is an association between androgens and PD in men, and animal models could be useful to evaluate this aspect. Up to date, there is a controversy about the effects of testosterone supplements on dopaminergic function, since both neuroprotective and toxic effects of testosterone are reported [73]. Animal models are recently being used also in studies exploring gender differences in cognition. It was shown that in male rats reduced androgen levels provide powerful and highly selective protection against the negative effects of 6-hydroxydopamine on memory functions [74]. Similarly, in another bilateral nigrostriatal dopamine lesion model of early PD, bilateral injections of 6-hydroxydopamine made in the lateral caudate one day after and/or 28 days before behavioral testings produced deficits in working memory and other executive functions only in male rats with normal circulating androgen levels, whereas in rats where androgen were depleted there were no additional deficits [75]. Therefore, the role of androgens deserves to be further evaluated also in PD patients.

### 3.2. Genetic Factors

A well-validated genetic risk factor for PD is a mutation in the gene encoding glucocerebrosidase (GBA), with a prevalence of GBA mutations among PD patients of 5–15% [76]. A recent systematic review and meta-analysis clarified that there is a gender difference between female and male PD patients, with a higher prevalence of these mutations in PD women [77].

Over the past two decades, research recognized that also mutations in the gene encoding leucine-rich repeat kinase 2 (LRRK2) are a common risk factor in both monogenic and sporadic forms of Parkinson’s disease [78], and more and more LRRK2 variants have been detected in different cohorts of PD patients with different geographical or ethnic origins. Several studies suggest that LRRK2-associated PD mutations are more common among female PD patients [79]. More recent studies focused on the idea that genetics, sex and the interactions between them drive phenotype: in a cohort of PD patients that carried LRRK2 G2385R variant, gender-related phenotypic differences were observed, since women manifest a more benign disease course, being at lower risk of impairment in activities of daily living and autonomic dysfunction, but also at higher risk to development of mood disorders, as compared to men [80]; similarly, San Luciano et al. demonstrated that the most prevalent LRRK2 mutation, the G2019S, is associated with a similar phenotype between male and female PD patients, even if women had a worse score in part IV of MDS-Unified Parkinson’s Disease Rating Scale (MDS-UPDR IV) [81]. Different phenotypes influenced by sex and genetics could also relate to cognitive decline: males with APOE ε4 had a steeper rate of cognitive decline [2]. Moreover, genetics is a relevant topic of research for the determination of PD risk factor and PD protective factors, and can help to understand epidemiologic differences between sexes too. Mitochondrial haplogroups demonstrated a significantly protective effect for PD risk only in females [3]; *GAPDH gene* variants are associated with an increased PD risk in men [82], while small CGG expansion (41–54 repeats) in the fragile X mental retardation 1 (FMR1) gene, called FMR1 “gray zone” alleles (GZ), are a significant risk factor for parkinsonism, more prominent in men [83].

### 3.3. Biomarkers

High urate levels have been associated with lower PD risk and with a better PD prognosis, but the association is consistent among men and weaker among women [1]. In addition, the association varied significantly by age among women, showing a protective effect only in women above 70 years, when urate levels are comparable to those in men, suggesting that estrogens may predominate in determining the lower risk of PD among women [84], suggesting a sex-specific protective effect of uric acid (UA) on nigrostriatal dopaminergic neurons [4].

Serum homocysteine has been studied as another potential biomarker for PD progression, and, interestingly, such association showed a sex difference: elevated serum homocysteine levels are associated with a greater motor impairment in PD males and poorer cognitive performance in PD females [5]. In the study just mentioned, cognitive functions were measured using a test battery including Mini Mental State Examination, Auditory verbal learning test, Copy and Delayed recall of Rey-Osterrieth complex figure, Clock drawing test, Boston naming test, Verbal fluency tasks, Symbol digit modality test, Trail making test and Stroop color word interference test while the severity of motor symptoms was assessed using MDS-UPDRS part III.

Some recent studies, aimed to evaluate circulating lipids as potential PD biomarkers, also showed gender differences. N-acylphosphatidylethanolamines, generally referred to by the acronym NAPE, are a class of phospholipids present at low concentrations in cellular membranes. Particular circulating NAPE species serum levels are significantly lower in PD patients compared to healthy controls, with a stronger decrement in female PD patients [6]. Luca et al. evaluated the role of serum lipid fractions as potential biomarkers for cognitive decline in a large cohort of PD patients, finding a sex-specific different contribute of lipids fractions on cognitive performance in PD, i.e., the association between hypertriglyceridemia and executive dysfunction only in women [7].

Other biomarkers, such as cerebrospinal fluid beta-amyloid 1–42, total tau, phosphorylated tau and unphosphorylated total alpha-synuclein showed no significant differences according to gender as demonstrated in a study conducted on 361 PD patients. The mean age and age at PD onset of the patients were 61.4 ± 9.8 years and 59.6 ± 9.8 years, respectively, and 238 (66%) patients were men. The mean MDS-UPDRS past III score was 21.2 ± 8.9. [8].

### 3.4. Clinical Features

#### 3.4.1. Motor Symptoms

The first study that explored gender differences in PD motor symptoms dates back to 2007 and involved about 250 de novo PD patients, showing that: (1) age at onset was 2.1 years later in women (53.4 years) than in men (51.3 years); (2) women more often presented with tremor (67%) than men (48%); (3) women had a 16% higher striatal [123I]FP-CIT single photon emission computed tomography ([123I]FP-CIT) binding than men at symptom onset and throughout the course of PD. These results suggest that, in women, the development of symptomatic PD may be delayed by higher physiological striatal dopamine levels, possibly due to the activity of estrogens [85].

Recently, new data on gender differences in PD were obtained from the Parkinson’s Progression Markers Initiative (PPMI) database, an international, multiple-site, prospective, longitudinal cohort study. A 5-year longitudinal analysis of a subgroup of 423 patients (65.4% men) from the PPMI cohort showed: (a) a similar increase over time of MDS-UPDRS part III OFF scores in both sexes, with an increase over time of MDS-UPDRS part III ON scores only in men; (b) a significant increase over time in Levodopa equivalent dose (LED) in men as compared to women [9].

Furthermore, in a cross-sectional descriptive survey conducted on 141 community-dwelling PD patients (84 males and 57 females), males reported more rigidity, speech problems, sexual dysfunction, memory problems, and socializing problems than females [10].

Interestingly, a correlation between coffee consumption and motor symptoms in PD patients has been described and the impact of caffeine intake on PD risk and mortality appears to differ by gender. The results of a cross-sectional study on a cohort of 284 PD patients demonstrated that coffee drinkers had lower tremor scores when compared to non-coffee drinkers, and the coffee consumption was inversely related to tremor severity in a dose-dependent manner in de novo PD patients. These relationships were statistically significant only with rest tremor, and not with action tremor. The severity of tremor was measured using tremor scores of MDS-UPDRS part III. The effect of coffee consumption on tremor severity was gender-dependent, and was significant only in men [11].

Moreover, in a study conducted on 415 PD patients, women with PD had significantly lower scores in MDS-UPDRS part III total score and postural tremor item of MDS-UPDRS III compared to men with PD. No significant gender-related differences were found in scores related to other motor symptoms [12].

Just recently, the results of a cohort study conducted in China to investigate the risk factors of motor complications in female patients with PD and the correlation between the occurrence of motor complications and sex hormone levels were published. The data collected showed that female PD patients have a higher incidence of motor complications than males. About sex hormen levels, conflicting data were obtained regarding the correlation between levels of prolactin and wearing-off phenomenon in female PD patients but further research to clarify is needed [13].

#### 3.4.2. Non Motor Symptoms

There is currently a large amount of evidence available on the gender differences in spectrum and severity of non-motor symptoms (NMS) in PD patients, suggesting a possible sex-related effect.It is noteworthy that there are numerous scales for evaluating NMS in PD, and for this reason it can often be difficult to compare the available data.

The Fattori di Rischio Ambientali e Genetici Associati alla Malattia di Parkinson (FRAGAMP) study (a large multicenter case-control study carried out in Central-Southern Italy to evaluate the possible role of environmental and genetic factors in PD) showed that presence of NMS is strongly associated with male rather than female gender. In fact, male sex showed a strong positive association with all NMS compared to women, except for urinary disturbances. Probably, the presence of several NMS, in particular depression and cognitive impairment among PD men, is more strictly linked to the neurodegenerative processes related to PD [14].

A cross-sectional analysis conducted on 569 drug-naïve PD patients in China demonstrated that: (1) the frequencies of sleep/fatigue and mood/apathy were more prevalent in women; (2) the frequencies of urinary and sexual dysfunction (SD) were more prevalent in men; (3) female sex, disease duration, UPDRS III score, depression, and NMSS subcores including sleep/fatigue, mood/apathy, attention/memory, and gastrointestinal were the determinants of quality of life (QoL) in patients with drug-naïve PD. NMS are common and heterogeneous in untreated PD patients and are important determinants of decreased QoL in PD, also presenting differences between genders [15].

SD was subsequently investigated using The International Index of Erectile Function (IIEF)37 (which is a validated, multidimensional, self-report instrument widely used for male sexual dysfunction) in an observational study that included 203 patients (113 males and 90 females) affected by PD and living in three different Italian regions. The collected data confirmed the higher prevalence of SD in male than in female PD patients. In more detail, the authors demonstrated that: (1) men were significantly more affected by SD than women, but no difference in the severity of the dysfunction emerged between genders; (2) around 85% of PD patients had a stable couple relationship, and about 40% were satisfied with such a relationship. However, about 57% of the patients stated that the disease affected their sexual life, especially due to reduced sexual desire, and the frequency of sexual intercourses. As it was reasonable to expect, also in this cohort of patients there was an impact of NMS on the QoL in PD patients [16].

As for the influence of NMS on QoL, a recent study conducted on 122 PD patients found that female patients reported reduced QoL due to depression and pain in early PD stages (Hoehn & Yahr I-II), while worsening of QoL in advanced PD was reported only by male patients in relation to the deterioration of the cognitive domain [17].

Interestingly, gender differences have also been demonstrated on some poorly investigated aspects of life and care of PD patients. A study conducted on 85 patients (of whom 52% were women) showed that while a spouse or partner is the most likely individual to serve as a caregiver, homebound female PD patients were more likely to be single or widowed compared to men. They were also more likely to lack any caregiver [18].

Moreover, in a retrospective study aimed to describe a sex-specific patient journey in Dutch PD patients during the first 5 years after diagnosis, significant sex differences were described: in the Netherlands, female patients visit most of the healthcare professionals sooner after diagnosis and more frequently than men. In addition, PD-related complications occur earlier in women than in men. A relevant percentage of patients is admitted to nursing homes within 5 years after diagnosis; again, this happens more frequently in women [19].

Anxiety, depression, mobility, emotional well-being, social support and bodily discomfort and psychosocial functioning, assessed with specific validated scales, were significantly worse in female PD patients in an observational study conducted on 157 PD patients [20].

Pain is a frequently reported NMS in PD patients. The currently available evidence proves that pain in PD is more frequent in women rather than men. [21,22]. The mechanisms by which females generally suffer more than males can be due to several biological aspects already discussed in previous literature [86,87,88]. In summary, PD women have lower muscle mass compared to men and this is relevant in relation to the fact that the most frequently reported pain in the course of PD has musculoskeletal characteristics; moreover, PD mostly affect women in postmenopausal phase, when estrogen and progesterone levels decline determining an alteration of perception of pain; lastly, females have a more sophisticated notion of self, which results in a deeper attention to symptoms in comparison to males [87]. Since chronic pain and depression often coexist and since female gender is considered to be a risk factor for pain in PD, we could suggest that managing pain in women could possibly improve their quality of life.

### 3.5. Pharmacological Treatment

Medical treatment of PD includes levodopa, dopamine agonists, anticholinergics, monoamine oxidase inhibitors, catechol-o-methyl transferase (COMT) inhibitors, amantadine and several other pharmacologic agents. Treatments may differ according to the patient’s symptoms, age, and responses to specific drugs but the choice between them, to date, is still not gender-oriented. In the previous review from our group [1], it was already remarked that similar dopaminergic treatments were assigned to men and women with PD, without gender preference, even if we are now aware of different pharmacokinetics and different outcomes in men and women. Women present a significantly greater bioavailability of L-dopa [23,89] and also lower levodopa clearance levels [90]. Moreover, women are known to develop more frequently levodopa-induced dyskinesias [91] and have an increased risk of the so-called “brittle response” to levodopa, defined as a susceptibility to dyskinesia when treated with small amounts of levodopa [92].

A very recent multicentric study on L-DOPA-naive PD patients receiving for the first time L-DOPA/benserazide formulation showed some interesting gender related-differences in Levodopa pharmacokinetics that could help to explain gender-related differences in levodopa clinical response: female sex and body mass index significantly predicted Area Under the Curve (AUC) and maximum concentration (Cmax) and, stratifying by gender, body mass index (BMI) predicted half-life (t1/2) only in women [93].

There is a possible relation between different genotypes and the therapeutic response to levodopa, that seems to be affected by sex differences. Sampaio et al. observed that carriers of monoamine oxidase B (MAO-B) (rs1799836) A and AA genotypes and COMT (rs4680) LL genotype suffered more frequently from levodopa-induced-dyskinesia, but also that there is an increased risk of 2.84-fold for male individuals carrying the MAO-B G allele to develop motor complications when treated with higher doses of levodopa (*p* = 0.04) [23]. Interestingly, MAO-B encoding gene is located on chromosome X, supporting the hypothesis of the existence of a different dopamine metabolism in men and women due to sexual dimorphism.

Studies investigating the impact of gender on medical treatments in PD are mainly focused on levodopa treatment, with poor evidence available on other antiparkinsonian drugs. Pellecchia et al. analyzed gender differences in clinical responses to treatment with safinamide [24], an antiparkinsonian drug with a peculiar mechanism of action since it acts both as a highly selective and reversible MAO-B inhibitor, and as a blocker of voltage-dependent Na+ and Ca2+ channels and inhibitor of stimulated glutamate release, targeting both dopaminergic and glutamatergic systems [94]. The study revealed higher prevalence of dyskinesia in women compared with males at baseline, with a similar reduction of patients with dyskinesia in both genders (up to 30%) over a 1-year follow-up; the prevalence of any fluctuations was similarly reduced in both genders after safinamide introduction; no gender differences in SAEs were detected [24].

UA levels modification represents another relevant topic to gender-oriented treatment of PD. Higher serum UA levels are associated with a slower disease progression of PD with a clear difference between male and female patients, since this association is markedly stronger in men than in women [95]. Starting from this evidence, Schwarzschild et al. investigated whether women and men with PD differ in their biochemical and clinical responses to long-term treatment with inosine in The Safety of Urate Elevation in Parkinson’s Disease (SURE-PD) trial, a randomized, double-blind, placebo-controlled, dose-finding trial of the urate precursor inosine. The study demonstrated potential sex differences with inosine producing greater serum and CSF urate elevations and a slower progression of PD in women than in men [25]. Even if further clinical trials of inosine in both genders are needed and planned in the ongoing SURE-PD3 trial (NCT02642393), these findings encourage consideration of gender-specific therapies of PD.

Lastly, gender differences need to be studied in symptomatic therapies for non motor symptoms too. For instance, PD female patients report pain more frequently than male patients [17,96] and identify it as a determinant of poorer quality of life [96], so we can speculate that females should receive more frequently pain treatment, but, to date, no difference in the frequency or type of pain treatment according to gender is reported in the literature.

### 3.6. Surgical Treatment

Deep brain stimulation (DBS) of the globus pallidus internus (GPi) or subthalamic nucleus (STN) is an accepted treatment for advanced PD when symptoms are no longer managed adequately with medications. There is a gender discrepancy in DBS utilization: female gender is underrepresented among PD patients treated with these surgical procedures, and also women access later to DBS, despite they tend to be more dyskinetic and present more motor fluctuations [1]. Also, primary indications for DBS in PD patients differ by sex, being medication refractory tremor a significative more common indication in males [26]. Speaking of outcomes, no significant differences between genders in clinical outcomes are reported. A recent Italian retrospective study confirms a similar clinical improvement of motor symptoms in males and females, even with some slight differences in the long-term follow-up, as a poorer improvement of bradykinesia at 1-year follow-up and non-dopaminergic features at 10 years follow-up in female patients [27].

In a very recent cross-sectional and longitudinal, prospective, observational, controlled, quasi-experimental, international multicenter study conducted on 505 PD patients treated with DBS, several evidence of a gender gap in DBS for PD emerges: fewer women underwent DBS indication assessments than to be expected from the gender ratio of the general PD population; preoperatively, mean PD duration was longer and dyskinesia more severe in women with PD; DBS was equally clinically efficacious on total QoL, nonmotor, and motor symptoms burden in women and men with PD [97]. Although these data are informative, they do not clarify the underlying reasons for gender disparities outlined here; therefore, further studies are needed to explore this topic.

The existence of gender differences in the impact of DBS on health-related quality of life (HRQoL) is controversial, since some studies reported higher improvement in men [28], and some others in women [98].

## 4. Conclusions

The impact of sex- and gender-related features on neurodegenerative disorders is still far from being fully elucidated. First, it’s important to realize that most of the discussed biological differences between men and women are both sex-related (i.e., genetic, hormonal, reproductive and physical) and gender-related (i.e., environmental influences such as society, culture or history). Moreover, future research should be oriented to uncover the biological basis and the physiopathological mechanisms mediating gender differences in PD. Focus on sex differences should be kept from pre-clinical (e.g., inclusion of female animals and consideration of hormonal states) to clinical studies, since it is more and more evident the necessity of targeted sex-oriented therapies. Similarly, observational, longitudinal post-marketing surveillance studies should also be analyzed with the aim to assess if and how sex and/or gender may affect the effectiveness and/or safety and tolerability of PD medications, as well as the existence of different risks factor and different clinical courses. With regard to the issue of gender differences in antiparkinsonian treatments, our paper reports new evidence upon levodopa and safinamide. The need for personalized medicine according to gender is now generally recognized, but to date no gender-oriented advice is available for PD. There is a lack of data on gender differences in response to antiparkinsonian drugs and adverse events. Future studies are therefore needed to develop gender tailored management in PD. To date, preclinical models of PD suggest that estrogens are neuroprotective, and also clinical studies support this evidence, since women display a less severe PD phenotype than men at presentation, and also severity of PD increases in post-menopausal women compared to pre-menopausal women. Likely, also genetic, epigenetic, sociocultural factors are involved in the complex relationship between gender and neurodegenerative disorders: for example, gender-differences in the quality of life are also related to social factors (such as cultural differences in gender roles). The overall goal should be encouraging clinicians to have special consideration for gender characterization and sex-specific issues in PD.

## Figures and Tables

**Table 1 brainsci-12-00917-t001:** Detailed results from studies on gender differences in PD reported in the paper from 2017 to present.

Study	Type of Study	Aim(s) of the Study	Number of PD Patient (Men%)	Mean AGE (Years)	Mean PD Duration (Years/Months)	Main Findings
Lee [2]	Case-control study (South Korea)	To evaluate a gender-sensitive resting-state networks (RSN) according to the uric acid level in drug naïve de novo patients with PD to elucidate the role of antioxidant in cortical functional networks of PD.	PD 90 (45.5%) HC 45 (44.4%)	Men PD-L-UA 64.8 ± 5.4 Men PD-H-UA 69.1 ± 10.1 Men HC 67.7 ± 8.3 Women PD-L-UA 68.3 ± 7.3 Women PD-H-UA 70.5 ± 1.3 Women HC 68.5 ± 6.9	Men PD-L-UA (low Uric Acid) 14.5 ± 9.4 months Men PD-H-UA (high Uric Acid) 21.1 ±19.6 months Women PD-L-UA 21.7 ± 21.0 months Women PD-H-UA 21.5 ± 18.6 months	(1) Interaction analysis showed a significant interaction in FC between PD-L-UA and PD-H- UA according to gender within the DAN, ECN, and DMN. (2) Compared to the control subjects, male patients with PD-H-UA had higher cortical FC, while female patients had lower cortical FC regardless of UA level within all seeds. (3) In a direct comparison, male patients with PD-H-UA had increased FC than did those with PD-L-UA. On the contrary, there was no significant difference in FC between PD-L-UA and PD-H-UA in female PD patients. These data suggest that resting-state FC might be closely and gender-specifically associated with the status of serum UA in de novo PD patients.
Loesch [3]	Clinic based cohort study (Australia)	To investigate whether the small CGG expansion (41–54 repeats)-FMR1 ‘grey zone’ alleles (GZ) contribute to the risk of parkinsonism in females as in males.	PD 601 HC 1005	NA	NA	The results provide evidence for a significant role of FMR1 GZ alleles as a risk factor for parkinsonism in females.
Baik [4]	Cross-sectional study (South Korea)	To investigate the relationship between the serum urate (UA) levels and patterns of striatal dopamine depletion in patients with de novo PD.	167 (49.70%)	Total 69.30 ± 9.82 Men 70.19 ± 9.68 Women 68.41 ± 9.93	Total 1.83 ±1.95 years Men 1.55 ± 1.31 years Women 2.09 ± 2.41 years	(1) Interaction analysis showed a significant interaction effect of sex and serum UA levels on the ISR in patients with PD. (2) In correlation analysis, the serum UA levels showed a significant inverse association with the ISR in all PD patients and particularly in male patients. (3) The serum UA levels were significantly associated with posterior putaminal DAT availability only in male PD patients. These data suggest the potentially close sex-specific relationship between the serum UA levels and ISR in patients with PD.
Yang [5]	Case-control study (China)	To examine the cognitive function of non-demented Parkinson’s disease patients and compare the results between male and female patients as well as control groups in search of any gender effect.	60 (50%)	Total 59.05 ± 9.55 Men 58.67 ± 10.23 Women 59.43 ± 8.98	Total 4.22 ± 5.33 years Men 4.37 ± 4.85 years Women 4.07 ± 5.85 years	There is a disparity between male and female patients in two domains of cognition. Male patients surpassed female patients on BNT (Boston Naming Test), a measure less commonly used to assess frontal lobe dysfunction, while female patients were superior on verbal retrieval test, reflecting the impairment of hippocampus. Since no significant differences were observed in these two measures between male and female controls, it is reasonable to infer that gender-based differences existed in PD patients.
Heller [6]	Case control study (Germany)	To explore the interplay of gender factors on emotion recognition and processing in PD.	PD 51 (51.0%) HC 44 (56.8%)	PD men 63.9 ± 8.4 PD women 64.0 ± 10.0 HC men 62.6 ± 9.0 HC women 63.4 ± 10.1	NA	(1) No group or gender differences emerged regarding cognitive abilities. (2) Male but not female PD patients exhibited confined impairments in recognizing the emotion anger accompanied by diminished neural response to facial expressions (e.g., in the putamen and insula). (3) Fear recognition was positively related with estrogen levels in female patients
Luca [7]	Cross-sectional study (part of PACOS study, The PArkinson’s disease COgnitive impairment Study)	To investigate the presence of possible associations between serum lipids fractions and executive dysfunction in PD patients, also exploring the sex-specific contribute of lipids level on cognition.	348 (57.47%)	Total 66.5 ± 9.5 Men 66.5 ± 9.7 Women 66.7 ± 9.2	Total 3.9 ± 4.9 years Men 3.4 ± 4.7 years Women 4.8 ± 5.0 years	(1) Only in women, a positive association between executive dysfunction and hypertriglyceridemia was found. Similarly, a negative correlation between triglycerides and FAB score was found only in women. (2) Among men, an inverse association was found between hypercholesterolemia and normal FAB performance.
Kim, 2 [8]	Data obtained from the PPMI database	To compare the longitudinal trajectories of cognition according to the presence of the apolipoprotein E (APOE) 4 allele in male and female PD patients.	361 (65.9%)	61.4 ± 9.8	6.8 ± 6.6 months	(1) Males with APOE4 had a steeper rate of cognitive decline than those without APOE4. In contrast, there was no significant interaction between APOE4 and time on longitudinal MoCA performance in the females. The main effect of APOE4 on the change in the MoCA score was not significant for either men or women. (2) When the data from both men and women were used, the APOE 4 +/M group exhibited a steeper rate of cognitive decline than did the APOE 4 +/F and APOE 4-/F groups.
Picillo, 1 [9]	Longitudinal study (data obtained from the PPMI database)	To examine sex-related longitudinal changes in motor and non-motor features and biologic biomarkers in early PD.	423 (64.5%)	Men 62.2 ± 9.7 Women 60.7 ± 9.6	Men 6.4 ± 5.9 months Women 7.1 ± 7.4 months	(1) Men experienced greater longitudinal decline in self-reported motor and non-motor aspects of experiences of daily living. (2) Compared to women, men had more longitudinal progression in clinician-assessed motor features in the ON medication state and required higher dopaminergic medication dosages over time. (3) Time to reach specific disease milestones and longitudinal changes in CSF biomarkers and DaTscan uptake were not different by sex.
Shin [10]	Cross-sectional descriptive survey (USA)	To compare self-reported motor and non-motor symptoms of PD by sex and disease duration.	141 participants (59.6%)	69.7 years	6.34 years	Males reported more rigidity, speech problems, sexual dysfunction, memory problems, and socializing problems than females.
Cho [11]	Cohort study (South Korea)	To assess the association between coffee consumption and motor symptoms in de novo PD patients based on their gender.	284 (51.76%)	65.76 ± 9.63	21.98 ± 26.13 months	(1) Coffee drinkers have lower tremor scores than non-coffee drinkers; (2) lower tremor scores in coffee drinkers are found in both the male and female subgroups; (3) coffee consumption is related to tremor in a dose-dependent manner; (4) the relationship between coffee consumption and tremor was statistically significant only in rest tremor, not in action tremor; (5) the dose-dependent, inverse relationship between coffee consumption and tremor scores was significant only in the male subgroup.
Kang [12]	Observational study (South Korea)	To investigate gender differences in clinical characteristics in patients with early PD.	415 (48.4%)	65.6 ± 9.5	17.1 ± 14.4 months	(1) Women with PD had a shorter duration of formal education than men with PD. (2) Women with PD had significantly lower scores in Unified Parkinson Disease Rating Scale part III and postural tremor compared to men with PD, which was significant after controlling for formal education. (3) Concerning non-motor symptoms, men with PD had higher scores of sexual function on the Non-Motor Symptoms Scale, which means sexual dysfunction was more severe or occurred more frequently in men with PD. Women with PD had significantly higher scores of sleep disturbance in the Pittsburgh Sleep Quality Index, which was not significant after adjustment for multiple comparison.
Wan [13]	Clinic-based cohort study (China)	To investigate the risk factors of motor complications in female patients with PD and the correlation between the occurrence of motor complications and sex hormone levels.	103 female PD patients	66.5 ± 10.2	4.0 ± 2.0 years	(1) Female PD patients have a higher incidence of motor complications. (2) Younger age of onset and higher H&Y stage were the risk factors for wearing-off phenomenon, and younger onset age was the risk factor for dyskinesia in female PD patients.
Nicoletti [14]	Large multicenter case-control study (Italy)	To evaluate the burden of non-motor symptoms (NMS) in PD and the possible gender differences in their occurrence.	585 (59.5%)	66.8 ± 9.8	7.2 ± 5.6 years	(1) PD women showed a significantly higher frequency of depression and urinary disturbances than parkinsonian men; a close frequency among PD women and men was recorded for hallucination, cognitive impairment, and sleep disorders. (2) PD men showed a stronger positive significant association with almost all NMS compared to women, excepting for urinary disturbances. The strongest association among PD men was recorded for cognitive impairment and depression.
Hu [15]	Cross-sectional analysis (China)	To explore the gender and onset age-related differences of non-motor symptoms (NMS) and the determinants of quality of life (QoL) in a large cohort of Chinese drug-naïve Parkinson’s disease (PD) patients.	569 (48.3%)	Total 58.1 ± 12.4 Men 57.9 ± 13.7 Women 58.2 ± 11.0	2.0 ± 1.5 years	(1) NMS was common in untreated PD patients, and the NMS profile was heterogeneous between different gender and onset age group. (2) NMS, especially for sleep/fatigue, mood/apathy, attention/memory, and gastrointestinal symptoms, were associated with the decreased QoL in patients with de novo PD.
Raciti [16]	Multicenter cross-sectional study (Italy)	To investigate prevalence of sexual disfunction and its variables, including gender differences, in PD patients.	203 (55,67%)	68.36 ± 8.5	7.78 ± 5.77 years	(1) Sexual dysfunction (SD) was observed in about 68% of men, and in around 53% of women, loss of libido being the main sexual concern in both sexes. (2) Men were significantly more affected by SD than women, but no difference in the severity of the dysfunction emerged between genders.
Balash [17]	Cross-sectional study (Israel)	To establish changes in self-assessment of the quality of life of patients with PD and their CGs, depending on their gender, in the early and late stages of the disease.	319 (64.57%)	68.3 ± 10.6	NA	(1) Significant differences on QoL between men and women in terms of emotional condition and pain perception, where women were more prone to depression and more sensitive to pain. (2) In advanced PD stages (H&Y III–V), males more often complained of memory decline and their QoL was significantly worsened in cognitive domain. (3) Male PD patients regarded their QoL better than female patients both in early disease and later in the advanced stages. This was largely due to the contribution of mobility items as well as of emotional items and pain, all of which had a greater effect in women. (4) On the other hand, cognition and communication contribute to worsened QoL more in men than in women. These differences persisted both in the early and more advanced stages of the disease. As the PD progressed from H&Y stages I-II to III-V, the QoL worsened both in male and female patients, primarily related to mobility domain.
Nwabuobi [18]	Cross-sectional study (USA)	To identify and describe differences in homebound men and women with advanced PD and related disorders, participating in an interdisciplinary home visit program.	85 (48%)	79.6	NA	The study showed that while a spouse or partner is the most likely individual to serve as a caregiver, homebound women were more likely to be single or widowed compared to men. They were also more likely to lack any caregiver.
Vlaanderen [19]	Retrospective study (Netherland)	To reconstruct a sex-specific patient journey for Dutch persons with PD during the first 5 years after diagnosis.	22293 (60.6%)	Males 71.6 ± 9.9 Females 72.5 ± 10.2	NA	(1) In the Netherlands, women visit most of the included healthcare professionals sooner after diagnosis and more frequently. (2) In addition, PD-related complications occur earlier in women than in men. (3) A sizeable subgroup of patients is admitted to nursing homes within 5 years after diagnosis. Again, this happens more frequently in women. (4) 14.6% of the women and 18.3% of the men died within 5 years after the diagnosis.
Farhadi [20]	Observational study (Iran/Sweden)	To determine independent sex differences in clinical manifestations and subtypes, psychosocial functioning, quality of life (QoL) and its domains between male and female individuals with PD.	157 (68.8%)	Men 61.4 Women 60.5	Men 6.1 years Women 8.2 years	Anxiety, depression, specific domains of HRQoL (mobility, emotional well-being, social support and bodily discomfort) and psychosocial functioning were significantly worse in female individuals with PD.
Zella [21]	Cross-sectional study (Germany)	To map pain in the largest PD study group to date; the analysis of the impact of different pain therapies in PD; to correlate the obtained results with gender.	2814 (57.2)	60.6 ± 8.1	9.9 ± 5.7 years	Excepted for the pain therapy with non-opioid analgesic drugs, the work did not demonstrate a significant association between the different pain treatments and gender.
Defazio [22]	Cross-sectional study (Italy)	To examine the association between pain and motor and non-motor factors in a large cohort of patients with PD.	Total PD 321 (40.80%) PD with pain 180 (50%) PD without pain (29.07%)	68.3 ± 9.2	Patients with pain 6.4 ± 4.4 years Patients without pain 6.7 ± 5.2 years	Pain was more frequent in women than men affected by PD.
Sampaio [23]	Clinic- based cohort study (Brazil)	To investigate the possible relation among selected single-nucleotide polymorphisms (SNPs) in the MAO-B (rs1799836) and COMT (rs4680) genes and the therapeutic response to levodopa (L-dopa).	162 (56.8%)	64.0 ± 9.4	7.4 ± 4.4 years	(1) Patients carrying MAO-B (rs1799836) A and AA genotypes and COMT (rs4680) LL genotype suffered more frequently from levodopa-induced-dyskinesia. (2) Male individuals carrying the MAO-B G allele had an increased risk of 2.84-fold of being treated with higher doses of levodopa.
Golfrè Andreasi [24]	Retrospective study (Italy)	To describe the long term-effects of STN-DBS on motor symptoms in a large cohort of PD patients with disabling motor complication; to describe separately the short-term and long-term outcomes of men and women patients, trying to discover any sex-related differences.	107 (66.35%)	Total at PD onset 43.35 ± 7.97 Men at PD onset 43.38 ± 8.38 Women at PD onset 43.28 ± 7.22 Total at surgery 54.60 ± 6.36 Men at surgery 54.52 ± 6.43 Women at surgery 54.75 ± 6.30	Total at surgery 11.31 ± 4.69 years Men at surgery 11.24 ± 4.89 years Women at surgery 11.44 ± 4.34 years	(1) No major differences in the motor outcome of STN-DBS between men and women. (2) Greater severity of dyskinesia in women, while the presence of motor fluctuations was comparable between the two sexes.
Schwarzschild [25]	Data obtained from The Safety of Urate Elevation in Parkinson’s Disease (SURE-PD) trial	To investigate whether women and men with PD differ in their biochemical and clinical responses to long-term treatment with inosine.	75 (45.3%)	NA	NA	Inosine produced greater increases in serum and CSF urate in women compared to men in the SURE-PD trial.
Solla [26]	Case-control study (Italy)	To assess the presence of specifc sex-related patterns in olfactory dysfunctions excluding the possibility of confounding effects in PD patients.	PD 99 (57.6%) HC 69 (44.9%)	PD women 69.6 ± 8.1 PD men 68.5 ± 7.5 HC women 69.2 ± 8.8 HC men 68.4 ± 9.9	3.6 ± 3.1 years	The PD male patients scored significantly lower on odor discrimination, identification, and Threshold-Discrimination-Identification score than females.
Abraham [27]	Secondary analysis of a cohort of PD patients seen at a tertiary care center (USA)	To examine differences in presentation, physician- and patient-reported PD outcomes, and progression by sex in a large clinical cohort.	1463 (62.4%)	Men 63.1 Women 63.4	NA	Females had significantly less social support, more psychological distress, and worse self-reported (but not physician-reported) disability and HRQoL at initial PD care visits, compared to mawles.
Nishikawa [28]	Multicenter, longitudinal, clinic-based study + cross sectional study	To identify the baseline differences between men and women, in terms of disease presentation, and to identify the influences of sex on longitudinal symptom trajectory.	Data from 12 longitudinal cohort (https://www.ncbi.nlm.nih.gov/pmc/articles/PMC7883324/table/T1/?report=objectonly) Accessed on 25 May 2022	Data from 12 longitudinal cohort (https://www.ncbi.nlm.nih.gov/pmc/articles/PMC7883324/table/T1/?report=objectonly) Accessed on 25 May 2022	Data from 12 longitudinal cohort (https://www.ncbi.nlm.nih.gov/pmc/articles/PMC7883324/table/T1/?report=objectonly) Accessed on 25 May 2022	(1) Female PD patients had a higher risk of developing dyskinesia early during the follow-up period, with a slower progression in activities of daily living difficulties, and a lower risk of developing cognitive impairments compared with male patients.
Zhu [29]	Large cross-sectional study (China)	To estimate the prevalence and identify potential risk factors influencing depression in non demented PD patients.	519 (62.8%)	65.35 ± 10.19	5.62 ± 4.19 years	(1) The most prevalent depressive domain was retardation (84.4%); global depression positively correlated with female sex. (2) Non-motor symptoms, poor sleep quality, younger age, and cognitive dysfunction are independent predictors of depression. Among these, non-motor symptoms or sleep disturbances are the most powerful predictors of each depressive domain.
Rocha [30]	Retrospective study (Portugal)	To analyse mortality in PD patients treated with DBS.	346 (60%)	60 ± 7 (mean age at surgery)	14 ± 6 years	The main causes of death were pneumonia, dementia, and acute myocardial infarction. In these series, male gender and disease duration until surgery were the only predictors of mortality in multivariate analysis.
Gee [31]	Longitudinal, prospective study (Canada)	To test the relationship between magnitude and spatial extent of atrophy in PD patients with progressive, significant cognitive decline and dementia (PDD).	33 (57.6%)	70.1 ± 3.3	8.4 ± 4.3 years	(1) More males developed PDD (7 males, 3 females) compared to those remaining intact (12 males, 11 females). (2) Clusters of lower grey matter volume were found in PDD compared to PD in left uncus at baseline and an expanded region that included the left hippocampus and parahippocampal gyrus at 36 months. The cognitive status by scan interaction showed differential changes between groups in the right insula.
Kim, 1 [32]	5 years prospective study (South Korea)	To assess the influence of sex on the short-term and long-term effects of subthalamic nucleus stimulation (STN-DBS) in Parkinson’s disease (PD).	100 (48%)	Men 57.3 ± 8.5 Women 60.2 ± 6.7	Men 10.8 ± 4.0 years Women 12.0 ± 4.7 years	(1) None of the changes from baseline to the 1- or 5-year follow-up in clinical outcomes differed between the men and women except for the 36-Item Short Form Health Survey (SF-36), which consists of physical-component summary (PCS) and mental-component summary scores. (2) Compared with baseline, there was an improvement in the PCS scores in both men and women at the 1-year follow-up; however, a trend toward greater improvement in men was observed. (3) At the 5-year follow-up, STN-DBS improved the PCS scores in men but not in women compared with the baseline, and there was a significant difference between the groups.
Oltra,1 [33]	Data obtained from the PPMI database	To investigate sex differences in brain atrophy and cognition in de novo PD patients.	PD 205 (61,9%) HC 69 (58%)	Men PD 63.80 ± 7.24 Women PD 61.76 ± 7.50 Men HC 64.05 ± 7.11 Women HC 60.55 ± 5.86	Men PD 62.84 ± 7.10 months Women PD 60.81 ± 7.52 months	(1) PD males had a greater motor and rapid eye movement sleep behavior disorder symptomatology than PD females. (2) PD males showed cortical thinning in postcentral and precentral regions, greater global cortical and subcortical atrophy, and smaller volumes in thalamus, caudate, putamen, pallidum, hippocampus, and brainstem, compared with PD females. Healthy controls only showed reduced hippocampal volume in males compared to females. (3) PD males performed worse than PD females in global cognition, immediate verbal recall, and mental processing speed. In both groups males performed worse than females in semantic verbal fluency and delayed verbal recall; females performed worse than males in visuospatial function.
Oltra, 2 [34]	Data obtained from the PPMI database	To investigate sex differences in cognition and brain atrophy in PD patients with and without probable RBD (pRBD).	Total PD 205 (61.95%) PD-non pRBD 126 (57.93%) PD-pRBD 79 (68.35%) HC 69 (57.97%)	Female PD-non pRBD 60.9 ± 7.4 Male PD-non pRBD 63.2 ± 7.4 Female PD-pRBD 63.5 ± 7.5 Male PD-pRBD 64.7 ± 7.0 Female HC 60.6 ± 5.9 Male HC 64.1 ± 7.1	Female PD-non pRBD 10.8 ± 8.5 years Male PD-non pRBD 10.3 ± 6.2 years Female PD-pRBD 9.9 ± 6.8 years Male PD-pRBD 11.6 ± 7.2 years	(1) Among drug-naïve patients, in the PD-pRBD group males had more severe motor and RBD symptomatology, worse cognitive performance, and greater subcortical volume atrophy than females. Such sex differences were also observed in subcortical volumes in PD-non pRBD group, but to a greater extent in the former. (2) Males in the PD-pRBD group had greater motor impairment and more RBD symptoms. (3) Cognitive impairment was also greater in males in the PD-pRBD group. Males performed significantly worse than females in MoCA, phonemic fluency and SDMT in the PD-pRBD group. By contrast, females in the PD-non pRBD showed greater impairment only in one semantic fluency test than males. (4) Global MRI measures revealed smaller total cortical and subcortical GM volumes in males of the PD-pRBD group, but not in the PD-non pRBD and control groups. In addition to the greater global atrophy found in males compared with females, differential volumetric atrophy according to sex was found in various subcortical structures. There was increased subcortical atrophy in males compared with females in both PD groups, and sex differences in subcortical regions were more evident in the PD-pRBD group, with a significant group-by-sex interaction in the pallidum. The sex effect in the pallidum was greater in the PD-pRBD group compared with the PD-non pRBD group.
Boccalini [35]	Data obtained from the PPMI database	To investigate gender influence on clinical features, dopaminergic dysfunction, and connectivity in patients with de novo idiopathic PD stratified according to the clinical criteria for subtypes (i.e., mild motor, intermediate, and diffuse-malignant)	286 (66.08%)	Males 62.32 ± 9.69 Females 61.35 ± 9.65	Males 2.05 ± 2.15 years Females 2.14 ± 2.34 years	(1) In the mild motor and intermediate subtypes, male PD patients had poorer cognitive abilities than females, who in contrast presented with more severe anxiety symptoms. (2) Gender differences emerged also in the diffuse malignant subtype, with more severe motor impairment in males than females, the former with associated more severe SUVr in the putamen. (3) The higher male vulnerability of the dopaminergic motor system also emerged in the mild motor and intermediate subtypes, with more severe and widespread connectivity alterations in the nigrostriatal dopaminergic pathway in males than females.
Shpiner [36]	Retrospective study (data collected from the University of Miami DBS Database)	To determine whether there is a gender disparity in patients undergoing deep brain stimulation (DBS) surgery for Parkinson’s disease (PD) at a single health system, and better understand the reasons for this discrepancy.	207 (75.8%)	NA	NA	(1) Female PD patients referred for DBS were significantly more likely to decide against DBS surgery based on their own preference. (2) Men were more likely not to go on for surgery due to loss to follow-up. (3) There were no significant differences in postsurgical motor outcomes as measured by MDS-UPDRS part III scores or need for PD medications, as measured by LEDD, when comparing men and women in our sample.
Perrin [37]	Cross-sectional study (Canada)	To determine how to differentiate more effectively depressed from non-depressed PD patients using the BDI and to determine which, if any, factors were different between men and women with depression.	307 (61.6%)	NA	4.53 years	Women were significantly more likely to be depressed than men.
Seyfried [38]	Cohort study (USA)	To present neurochemical data on the content and composition of glycolipids, phospholipids and cholesterol in SN tissue from male and female PD subjects and age-matched controls.	PD 12 (58. 33%) HC 18 (55.55%)	Male HC 75.6 ± 2.4 Male PD 78.4 ± 3.5 Female HC 72.6 ± 1.7 Female PD 75.8 ± 3.5	Male PD 16.2 ± 3.3 years Female PD 6.8 ± 2.2 years	(1) Disease duration before death was noticeably less for the female PD patients than for the male PD patients. (2) A highly significant correlation was found between total SN ganglioside content (lg/100 mg dry weight) and SN water content both within sexes and across sexes.
Dalrymple [39]	Clinic-based cohort study (USA)	To investigate whether the characteristics of PD patients differ based on the primary indication for deep brain stimulation (DBS).	137 (69.3%)	63.3 ± 7.6	10.1 ± 4.6 years	29 (93.5%) of 31 PD patients who underwent DBS for medication refractory tremor were men, and 66 (62.3%) of 106 PD patients who underwent DBS for motor fluctuations were men.
Wang [40]	Clinic-based cohort interview (USA)	To identify reliable emotional cues from expressive behavior in women and men with PD.	96 (72.9%)	66.46 ± 9.07	6.97 ± 5.36 months	(1) More gross motor expressivity and smiling/laughing indicated more positive affect in the total sample. Less conversational engagement indicated more negative affect in women. However, women with more negative affect and depression appeared to smile and laugh more.
Hamid [41]	Cross-sectional study (Rome)	To evaluate the potential of a particular set of N-acyl-phosphatidylethanolamines (NAPEs) as potential non-invasive plasma markers of ongoing neurodegeneration from PD in human subjects.	PD 177 (85.31%) HC 177 (80.22%)	HC 52 PD 63	NA	(1) The down-regulation of particular circulating NAPE species might have the potential to become a candidate biomarker for PD in female subjects. (2) NAPEs are significantly downregulated in the plasma of PD patients, with gender-specific profiles.
De Micco [42]	Case control study (Italy)	To investigate the potential sex-difference effect on the spontaneous neuronal activity within the most reported resting-state networks in early untreated PD patients and its correlation with baseline and longitudinal clinical features in PD criteria for development of PD dementia and compared them with level II (comprehensive) criteria.	PD 56 (53.5%)	Male HC 60.1 Female HC 55.7 Male PD 60.2 Female PD 58.5	Male PD 1.3 ± 0.7 years Female PD 1.5 ± 0.5 years	(1) Compared to female PD patients and healthy controls, male PD patients showed an abnormal spectral composition of the sensorimotor and dorsal attention networks in the slow-5 band. (2) The region-of-interest analysis showed an increased connectivity within the basal ganglia in female PD patients compared to males. (3) Functional sensorimotor connectivity changes at baseline showed to be an independent predictor of disease severity at 2-year follow-up.
Pellecchia [43]	Multinational multicenter, observational study	To analyze gender differences on clinical effects of safinamide in PD patients treated in real-life conditions during the SYNAPSES trial.	1610 (61.73%)	Total PD 68.4 ± 9.7 PD men 67.8 ± 9.7 PD women 69.4 ± 9.4	Total PD 8.8 ± 5.6 years Male PD 8.8 ± 5.6 years Female PD 8.9 ± 5.5 years	(1) The development of symptomatic PD is slightly delayed in women compared with men. (2) Women showed a more severe postural instability and bradykinesia compared to men; rigidity was more severe in men, whereas the severity of tremor was similar. (3) The addition of safinamide improved all cardinal symptoms in both genders. (4) Higher prevalence of levodopa-induced dyskinesia and WO in women vs. men with PD.
Liu [44]	Cross-sectional study (China)	To explore the features of excessive daytime sleepiness (EDS) and night-time sleep quality in PD patients of different sexes and age at onset (AAO).	586 (59.21%)	Men 65.7 ± 10.2 Women 64.1 ± 9.03	Men 36.0 years Women 36.0 years	Male patients are more likely to have EDS than female patients.
Bakeberg [45]	Clinical based study (Australia)	To investigate elevated serum homocysteine levels and symptom progression in a cohort of PD patients.	PD 205 (62.43%) HC 78 (39.7%)	PD 64.0 ± 9.38	8.9 years	(1) PD patients displayed significantly elevated homocysteine levels. (2) A significant positive correlation between homocysteine and MDS-UPDRS III score was identified in males with PD, but not in females, whereas a significant negative correlation between homocysteine levels and total ACE-R score was observed in females with PD, but not in males. (3) Multivariate general linear models confirmed that homocysteine was significantly predictive of MDS-UPDRS III score in male patients and predictive of total ACE-R score in female patients.
Georgiou [46]	Case-control study (Cyprus)	To investigate the presence of associations between mtDNA haplogroups and the risk for PD in the Cypriot population.	PD 230 (5.5%) HC 457 (50%)	PD 66.5 ± 10.5 HC 65 ± 10.7	NA	(1) Statistically significant associations regarding PD risk and PD age of onset were mostly detected for females thus suggesting that gender is a risk modifier between mitochondrial haplogroups and PD status/PD age of onset.
Fullard [47]	Retrospective cohort study (USA)	To examine sex differences and trends in comorbid disease and health care utilization in individuals with newly diagnosed PD.	133 (47.1%)	NA	Newly diagnosed	(1) Female PD patients had higher adjusted incidence rate ratio (IRR) of depression, hip fracture, osteoporosis, and rheumatoid/osteoarthritis than men. (2) In spite of greater survival, women with PD used home health and skilled nursing facility care more often, and had less outpatient physician contact than men throughout the study period.
Cholerton [48]	Prospective cohort study (USA)	To identify whether prediction of cognitive progression is aided by examining baseline cognitive features and whether this differs according to stage of cognitive disease.	PD-MCI, Progressed 86 (76.7%) PD-MCI, Stable 248 (71.4%)	PD-MCI, Progressed 71.2 (9.1) PD-MCI, Stable 66.9 (8.0)	PD-MCI progressed 9.5 years PD-MCI stable 8.7 years	(1) Processing speed and working memory were associated with conversion to PDD among those with PD-MCI at baseline, over and above demographic variables. (2) Conversely, the primary predictive factor in the transition from no cognitive impairment to PD-MCI or PDD was male sex (OR = 4.47, *p* = 0.004), and males progressed more rapidly than females (*p* = 0.01). Further, among females with shorter disease duration, progression was slower than for their male counterparts, and poor baseline performance on semantic verbal fluency was associated with shorter time to cognitive impairment in females but not in males.
Wu [49]	Case-control study (China)	To compare sex differences in clinical features and related factors of underweight and BMI in Chinese de novo PD patients.	PD 253 (57.70%) HC 218 (50.45%)	Male PD 63.8 ±10.7 Female PD 62.0 ± 9.9 Male HC 65.5 ± 11.1 Female HC 64.8 ±11.7	Male PD 18.2 ± 14.1 years Female PD 17.9 ± 13.8 years	(1) The BMI levels of both male and female patients were lower than those of the healthy population, and female patients had higher TC, TG, LDL-C, HAMD, and BMI values and a lower incidence of underweight and HCY levels than male patients. (2) Correlations between underweight and Δ SBP, Δ DBP, and UPDRS motor scores were found only in male patients. (3) BMI was correlated with ΔSBP and ΔDBP values in both sexes, and BMI was also associated with MoCA and lower UPDRS motor scores in male patients but lower HAMD scores in female patients.
Picillo, 2 [50]	Prospective longitudinal study (subgroup of patients from PRIAMO study, Italy)	To demonstrate that the presence of active sexual life is associated with disease progression in early PD.	355 (67.04%)	Female PD with active sexual life 61.27 ± 8.57 Female PD without active sexual life 68.06 ± 6.48 Male PD with active sexual life 62.51 ± 9.77 Male PD without active sexual life 70.84 ± 7.24	Female PD with active sexual life 4.89 ± 3.96 years Female PD without active sexual life 6.08 ± 8.19 years Male PD with active sexual life 5.1 ± 3.6 years Male PD without active sexual life 5.15 ± 4.4 years	Men were sexually active twice as much as women; sexually active men displayed distinctive demographic and clinical features compared to sexually non-active men.
Picillo, 3 [51]	Data obtained from PRIAMO study dataset (Italy)	To explore the impact of sex and age on relationship between prodromal constipation and disease phenotype in PD at early stages.	385 (64.15%)	Men without constipation 65.05 ± 9.57 Women without constipation 65.83 ± 8.82 Men with constipation 68.24 ± 8.96 Women with constipation 67.92 ± 7.57	Men without constipation 4.84 ± 3.75 years Women without constipation 5.88 ± 7.55 years Men with constipation 5.18 ± 4.41 years Women with constipation 5.16 ± 4.54 years	When examining the impact of sex, the presence of prodromal constipation was associated with attention/memory complaints and apathy as well as with a trend towards significance for lower cognitive performances in women only.
Porta [52]	Cross-sectional study (Italy)	To quantitatively investigate the existence of differences in spatiotemporal and kinematic parameters of gait in men and women with Parkinson disease (PD) using computerized 3-dimensional gait analysis.	35 (51.42%)	Women 70.8 ± 8.2 Men 70.7 ± 5.8	NA	PD is associated with specific sex-related modifications of gait, especially regarding ankle kinematics, while no differences were found in most spatiotemporal parameters, although men exhibited a wider step width. The only significant difference between men and women with PD in spatiotemporal parameters was a wider base of support in men.
Caplliure-Llopis [53]	Descriptive, quantitative and cross-sectional pilot study (Spain)	To describe the bone quality of a population with PD by calcaneal ultrasound and to compare it with a healthy control, assessing the influence of possible sex differences.	PD 21 (57.14%) HC 30 (43.33%)	Total PD 71.75 ± 3.7 Male PD 70.7 ± 8.6 Female PD 71.3 ± 3.7 Total HC 67.93 Male HC 67.9 ± 7.1 Female HC 68.5 ± 4.8	NA	Poorer bone quality in female patients with PD, who present a higher percentage of osteoporosis than healthy controls.
Kusters [54]	Two population-based studies (PASIDA study, Denmark and PEG study, United States).	To assess the association between age at menopause and age at menarche with PD risk.	805 female PD patients	68.4	NA	A later age at menopause was associated with a decreased risk of PD in women.
Kolmancic [55]	Clinical based study (Slovenia)	To assest gender differences in motor cortex measurements.	41 (53.7%)	Men 63.32 Women 64.22	Men 19.50 months Women 22.78 months	(1) Male patients had disturbed interhemispheric balance of motor thresholds, caused by decreased resting and active motor thresholds in the more affected hemisphere. Short interval intracortical inhibition was more effective in female compared to male patients in both hemispheres. (2) Female patients had a preserved physiological focal response to sensorimotor plasticity protocol, whereas male patients showed an abnormal spread of the protocol effect.
Savica [56]	Epidemiologic study (USA)	To project the prevalence of PD with and without dementia in the United States by 2060.	296 (63.2%)	NA	NA	The prevalence of PD was higher in men than in women for all subtypes and all age groups.
Tremblay [57]	Data obtained from the PPMI database	To compare brain features in Parkinson’s disease males and females using a multimodal imaging approach including deformation-based morphometry (DBM), cortical thickness and diffusion-weighted MRI measures in a large sample of patients with Parkinson’s disease at the early drug-naıve stage and healthy control participants.	232 (64.22%)	Men 61 ± 9 Women 60 ± 9	Men 7 ± 7 months Women 7 ± 7 months	Neurodegeneration and white matter damage are already more severe in males at the earliest symptomatic stage of the disease.
Meng [58]	Cross-sectional study (China)	To improve understanding of gender differences on quality of life (QoL) in PD patients of a different race, the differences of clinical features and health-related quality of life (HRQoL) between PD males and females were studied in a small cohort early to middle stage of Chinese PD patients.	162 (43.20%)	Men 60.41 ± 9.23 Women 59.60 ± 7.24	Men 6.56 ± 3.91 Women 6.38 ± 3.92	(1) Female patients have poorer QoL than male patients, especially bodily discomfort, stigma and emotional well-being. (2) Males had a higher RBDSQ score and lower HAMA and HRS scores compared to females.
Rusz [59]	Part of a longitudinal project “biomarkers in PD (BIO-PD)”	To estimate the occurrence and characteristics of speech disorder in early, drug- naive PD patients with relation to gender and dopamine transporter imaging.	100 (60%)	Men 60.8 ± 11.7 Women 61.2 ± 13.3	Men 1.9 ± 1.5 years Women 1.9 ± 1.6 years	Women showed better speech performance (*p* < 0.001) in dimensions reflecting voice quality (hoarseness and dysphonia/breathiness), consonant articulation (imprecise consonants), and pause production (prolonged pauses), whereas men were better (*p* < 0.001) only in loudness variability (monoloudness).
Sperens [60]	Retrospective, exploratory study (part of an ongoing longitudinal prospective population based project, the NYPUM project, New Parkinsonism in UMeå)	To explore the changes in the activities of daily living (ADL) in PD patients with, and to investigate possible differences in ADL performance between men and women with PD.	129 (58.91%)	Women 70.5 ± 9.8 Men 70.5 ± 10.2	Mean age at diagnosis: All 70.5 ± 10.0 years Men 70.5 ± 10.2 years Women 70.5 ± 9.8 years	Men and women had a similar self-assessed decrease of ADL-performance over time, with the exceptions of the activities “Shopping and Cleaning”, which were more demanding in women, a fact probably more due to a gender-related division in activity patterns than a PD-specific finding.
Yoon [61]	Cross-sectional study with control group (South Korea)	To identify whether there are gender differences of health-related quality of life (HRQoL) in PD patients in the early stages, and which NMSs are associated with HRQoL depending on gender.	PD 89 (52.80%) HC 36	Men PD 68.18 ± 8.14 Women PD 68.90 ± 7.71 HC 65.20 ± 10.76	Men PD 2.56 ± 2.79 years women PD 3.16 ± 3.63 years	(1) In the comparison of NMSs, PD patients showed higher scores in PDQ-39 compared to normal controls especially in female patients. HRQoL would be involved even in early stages of PD, especially in females. (2) Fatigue and depression were the main determinants of poor HRQoL in female patients even in early stages
Iwaki [62]	Clinic-based cohort study (Japan)	To examine the factors that affect levodopa pharmacokinetic in patients with PD.	220 (50.91%)	68.1 ± 8.9	7.7 ± 5.8 years	(1) Age, sex, and body weight were significantly related to AUC. Among these factors, female sex was the strongest contributing factor that increased AUC. (2) Based on the predictive model, it was assumed that the AUC of a woman receiving 300 mg of levodopa is equivalent to that of a man receiving 354 mg of levodopa.

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
