# Peer review of "Sex Differences in Parkinson’s Disease: From Bench to Bedside"

_brainsci, 2022, doi:10.3390/brainsci12070917_

Round 1
Reviewer 1 Report
Thank you for the opportunity to review this commentary, in which Russillo et al. discuss preclinical and clinical evidence on gender differences in pathophysiology, clinical presentation and therapeutic responses in Parkinson’s disease (PD). This is an important and interesting issue.
I would suggest following minor amendments to be considered by the authors:
1. Introduction
I suggest replacing the verb “collect” with “discuss” or “present”.
2. Methods
Please add what was the latest date of the publications included and also the criteria for the selection of the studies included.
3. “Results” is missing. The section should be further sub-divided into 3.1. Evidence from animal studies, 3.2. Genetic factors etc.
Acronyms should be only used for terms that appear in the text at least 5 times, otherwise please use full names. Full names need to be used always when the term appears for the first time.
Line 72 – 6-OHDA is misspelled.
Line 75 – “where” is a misspelling.
4. It is inappropriate to label one of the mutations “the most important”, perhaps this could be replaced with the most frequent?
Line 90 - *genetics
Line 95 - * higher risk of development
Line 102 - *between sexes
5. Biomarkers – please add how the motor and cognitive performances were measured
Line 130 – 133: Please provide the sample size, demographic and disease-specific features of the mentioned study.
6. I would suggest re-naming this into: 6. Clinical phenotype; 6.1. Motor Symptoms 6.2. Non-motor symptoms
For all the studies mentioned, please add the male:female ratio, mean age, mean disease duration and mean HY stage (if available).
Line 145 – please use the full name of the database
Line 151 – 154: please add demographic and disease specific details
Line 159 – how was the severity of tremor measured?
Line 165 – how was the postural tremor measured? If this is based on the UPDRS Part III assessment, please specify the item.
Line 168 – “clinical-based” cohort is an unusual term, could you, please, explain it?
7. Please use the full names for the studies, with acronyms in brackets.
Line 187 – 189: This is not fully clear. Could you, please, explain how was the exact pathophysiology of non-motor symptoms unearthed in this study?
Line 193 – Should this sentence be divided into 2?
Line 199 – how was SD assessed?
The notion that female patients reported reduced QoL due to depression and pain in early PD stages is important, particularly as female gender is considered to be a risk factor for pain in PD. Please comment on this.
Line 221 – how was this study conducted?
Line 228 – how were these symptoms assessed?
8. Line 256 – Acronym LD is used for the first time here, please. Also, please expand on acronyms AUC, Cmax, t1/2. Full names should be used.
Line 262 – Again, acronyms should not be used when the term appears for the first time.
9. Perhaps a recently published work on gender gap in DBS can be included - https://pubmed.ncbi.nlm.nih.gov/35444187/
Reviewer 2 Report
Taking into account the possible sex differences in Parkinson’s disease patients is of great importance concerning both their treatments and quality of life. Therefore, the topic of this review is significant and highly interesting.
There are only some minor comments:
The authors state that they summarize the research after the article of Picillo et al. 2017. However, the general reader may not be familiar with that review. Therefore, I would suggest to better compare/emphasize the conclusions, new observations of that review and this one.
Concerning the genetic factors, GBA and LRRK2 are mentioned. Are there data concerning the other genes (alpha-synuclein, PINK1, ...) ?
The introduction is almost the same as the abstract.
Please, pay attention to the references (the first mentioned are 62-65 in line 41) as well as abbreviations (what MPTP, OHDA, DOHA or MDS-UPDRS stand for?).
Reviewer 3 Report
Therapeutic advances in PD treatment could be expanded
Neurotoxic ramifications slightly more described
